# Forward Inverse Relaxation Model Incorporating Movement Duration Optimization

**DOI:** 10.3390/brainsci11020149

**Published:** 2021-01-23

**Authors:** Misaki Takeda, Isao Nambu, Yasuhiro Wada

**Affiliations:** Graduate School of Engineering, Nagaoka University of Technology, Nagaoka, Niigata 940-2188, Japan; inambu@vos.nagaokaut.ac.jp (I.N.); ywada@vos.nagaokaut.ac.jp (Y.W.)

**Keywords:** human arm movement, movement duration, reaching movement, arm dynamics, optimization model, forward inverse relaxation model, signal-dependent noise, speed-accuracy trade-off

## Abstract

A computational trajectory formation model based on the optimization principle, which introduces the forward inverse relaxation model (FIRM) as the hardware and algorithm, represents the features of human arm movements well. However, in this model, the movement duration was defined as a given value and not as a planned value. According to considerable empirical facts, movement duration changes depending on task factors, such as required accuracy and movement distance thus, it is considered that there are some criteria that optimize the cost function. Therefore, we propose a FIRM that incorporates a movement duration optimization module. The movement duration optimization module minimizes the weighted sum of the commanded torque change term as the trajectory cost, and the tolerance term as the cost of time. We conducted a behavioral experiment to examine how well the movement duration obtained by the model reproduces the true movement. The results suggested that the model movement duration was close to the true movement. In addition, the trajectory generated by inputting the obtained movement duration to the FIRM reproduced the features of the actual trajectory well. These findings verify the use of this computational model in measuring human arm movements.

## 1. Introduction

Toward elucidating information processing in the human brain, neuroscience research with a computational approach is being conducted. This approach understands the brain by considering its operating principles and expresses these processes with a mathematical model. If computational research advances the understanding of the mechanisms of the human brain, it will be possible to utilize the superior functions of the brain in various fields such as engineering, medical care, and rehabilitation.

David Marr, a pioneer in computational research, stated that three-level perspectives: Computational theory, representation and algorithm, and hardware implementation are important in understanding information processing in the human brain [1]. Computational theory research focusing on trajectory planning in human motor control, has been actively conducted, and several models for trajectory generation based on optimization criteria have been proposed since the end of the 20th century. The primary optimization criteria for generating human movement trajectory are the minimum jerk criterion [2], minimum commanded torque change criterion [3,4], minimum muscle-tension change criterion [5], minimum motor command change criterion [6], and minimum variance criterion [7]. It has been reported that the trajectory generated by the minimum commanded torque change criterion is the closest to the human trajectory among movement criteria that can be quantitatively evaluated [4,8,9]. Several neural network models have been proposed as representation, algorithm, and hardware that can satisfy these computational criteria [10,11,12,13]. Among them, Wada and Kawato’s computational model for trajectory formation based on the optimization principle is a model in which the minimum commanded torque change criterion is adopted as the computational theory, and the forward inverse relaxation model (FIRM) is introduced as the hardware and algorithm [14,15]. The FIRM can reproduce the features of human reaching movements as well as complicated human motion trajectories, such as handwriting [16,17]. The introduction of FIRM has overcome three problems of conventional neural network models: (1) The spatial representation of time is used, (2) back propagation is essential, and (3) too many iterations to obtain an optimal trajectory are required, by alternately and repeatedly calculating the forward and inverse dynamics models. The via-point representation of early FIRM had both spatial and temporal information, but with this mechanism, the time of passing through the via-point position could not be redetermined when the entire movement duration (the time from the start to the end) was changed. Therefore, Wada and Kawato proposed an algorithm that can optimally determine the via-point time without explicitly expressing the time information, using only spatial information as the via-point representation [15]. Consequently, the current FIRM has been improved such that the movement trajectory, the motor command torque, and the position and time of the via-point can be optimally determined using the entire movement duration and spatial conditions given as inputs.

However, considering actual human movement, the entire movement duration is not always given as an input, and as such, it may be optimally determined based on certain criteria [18,19]. According to empirical facts, including Fitts’ law [20,21], the movement duration changes depending on task factors, such as required accuracy and movement distance. Indeed, how the movement duration is determined is one of the major issues in computational studies of reaching movement.

In order to achieve accurate, smooth, and low-energy (i.e., low-exerted-torque) movement, it is advisable to lengthen the movement duration and move slowly. Tanaka et al. proposed a minimum time model [19], in which the shortest movement duration is determined under the constraint that endpoint accuracy meets a given tolerance in the presence of signal-dependent noise [7]. However, Mazzoni et al. pointed out that patients with Parkinson’s disease are more likely to choose to move slowly, even though they can move as fast as control subjects [22] thus, the minimum time criterion alone may not be sufficient. They also suggested that implicit motor motivation may be involved in determining movement duration, because Parkinson’s disease patients show symptoms such as bradykinesia, due to a decrease in dopamine, which is said to be involved in motor motivation. Moreover, Berret and Jean estimated the cost of time (CoT) from behavioral experimental data using the inverse optimal control approach, assuming that the CoT and the trajectory cost have an additive relationship [18]. The results of their study demonstrated a striking sigmoidal shape of the CoT for self-paced reaching.

The primary purpose of our study is to elucidate the information processing mechanism of the human brain, especially during movement duration planning. Hence, we propose a model for trajectory planning that incorporates movement duration planning into the FIRM that considers the smoothness of motor commands and endpoint tolerance. Although the FIRM is an existing model, the module of the movement duration planning algorithm is the novelty of this research. The movement durations, hand paths, and velocity profiles estimated by our model well represented the features of human reaching movement. This suggests that it may explain motor information processing in the human brain. Furthermore, we estimated the distributions of the difference between the mean of the measured movement duration and model movement duration using Bayesian statistics and investigated how much difference there was between them. Finally, we described the limitations of the model.

## 2. Movement Duration Optimization Module

The computational model of the point-to-point arm reaching movement consists of the movement duration optimization model proposed in this paper and the FIRM. A schematic diagram of this model is shown in Figure 1. The inputs of this model are the tolerance ($W^{⋆}$), which is the allowable error, such as the radius of the target, and the starting and target positions ($x_{s}$, $y_{s}$, $x_{f}$, $y_{f}$), which are given first as visual information. To find the movement duration *D* that minimizes the cost function *C*, it is necessary to calculate *a* and γ, which are determined based on the arm dynamics. To determine *a* and γ, the time-normalized trajectory is determined by the FIRM when $D^{\mathrm{tmp}}=1.0\;s$ is the input. The steepest descent method finds the optimal solution $D^{⋆}$ of the cost function *C* (see Section 3.4.5 for details of the engineering algorithm), and the optimal trajectory is generated by the FIRM using the $D^{⋆}$. Thus, the movement duration ($D^{⋆}$), commanded torque ($\tau^{⋆}\left(t\right)$), and trajectory ($\theta^{⋆}\left(t\right)$) are finally obtained as outputs.

We propose that the movement duration is determined by the optimization of the cost function which is expressed as a weighted sum of two terms related to smoothness and meet-tolerance as follows:(1)$$C=C_{\tau }+\lambda C_{\mathrm{err}}\to \mathrm{Min}.,$$
(2)$$\begin{array}{cc} C_{\tau } & =\sum_{i=1}^{N}\int_{0}^{D}{\left(\frac{d\tau_{i}\left(t\right)}{dt}\right)}^{2}dt, \end{array}$$
(3)$$\begin{array}{cc} C_{\mathrm{err}} & ={\left(W^{⋆}-\hat{W}\right)}^{2}, \end{array}$$
where $\tau_{i}\left(t\right)$ denotes the *i*–th joint commanded torque at time *t*, *t* changes from 0 to movement duration *D*, and *N* represents the number of joints. The smoothness term $C_{\tau }$ is the sum of the commanded torque change, and the tolerance term $C_{\mathrm{err}}$ is the squared error between the tolerance $W^{⋆}$, given as the task condition and the endpoint variability estimated in the brain $\hat{W}$. $C_{\mathrm{err}}$ takes the minimum when $W^{⋆}$ is equal to $\hat{W}$, but the whole cost function *C* should be minimized to consider trajectory smoothness. The weight λ indicates how much more important the tolerance term is than the smoothness term. Figure 2 shows a schematic diagram of this cost function. $C_{\tau }$ and $C_{\mathrm{err}}$ can be approximately expressed as a function of the movement duration *D*, as shown in the following equation: (4)$$\begin{array}{cc} C_{\tau } & ≃\frac{a}{D^{5}}, \end{array}$$(5)$$\begin{array}{cc} C_{\mathrm{err}} & ≃{\left(W^{⋆}-\frac{\gamma }{D}\right)}^{2}. \end{array}$$

The mathematical proof of these equations is given in the next section.

### 2.1. Proof of $C_{\tau }≃a/D^{5}$

The multi-joint human arm dynamics on the horizontal plane is as follows: [3]
(6)$$\tau \left(t\right)=M\left(\theta (t)\right)\ddot{\theta }\left(t\right)+h_{1}\left(\theta (t)\right)\left[\dot{\theta }\left(t\right)\dot{\theta }\left(t\right)\right]+h_{2}\left(\theta (t)\right)\left[{\dot{\theta }\left(t\right)}^{2}\right]+B\dot{\theta }\left(t\right),$$
where τ(t) is the joint torque vector, and θ(t), $\dot{\theta }\left(t\right)$, and $\ddot{\theta }\left(t\right)$ are the joint angular position, velocity, and acceleration, respectively. The first term is the inertial force given in terms of the inertia matrix $M\left(\theta (t)\right)$. The second term is the Coriolis force, which is given in terms of a Coriolis force coefficient matrix $h_{1}\left(\theta (t)\right)$. The third and fourth terms are the centrifugal and viscous forces in terms of the coefficient matrices of the centrifugal and viscous forces, $h_{2}\left(\theta (t)\right)$ and B, respectively. Additionally, $\left[\dot{\theta }\left(t\right)\dot{\theta }\left(t\right)\right]$ and $\left[{\dot{\theta }\left(t\right)}^{2}\right]$ are the vectors in the Coriolis and centrifugal forces, ${\left[{\dot{\theta }}_{1}\left(t\right){\dot{\theta }}_{2}\left(t\right),{\dot{\theta }}_{1}\left(t\right){\dot{\theta }}_{3}\left(t\right),\cdots ,{\dot{\theta }}_{N-1}\left(t\right){\dot{\theta }}_{N}\left(t\right)\right]}^{T}$ and ${\left[{{\dot{\theta }}_{1}\left(t\right)}^{2},{{\dot{\theta }}_{2}\left(t\right)}^{2},\cdots ,{{\dot{\theta }}_{N}\left(t\right)}^{2}\right]}^{T}$, respectively (the superscript T denotes the transpose of a vector or matrix).

The above equation can be written as a function of the movement duration *D*, by introducing the time-normalized variable *s* (0≤s≤1), as follows (see [23] for details):(7)$$\begin{array}{cc} \tau (t)=\tau (Ds) & ≃\frac{1}{D^{2}}\tilde{\tau }\left(s\right), \end{array}$$(8)$$\begin{array}{cc} \tilde{\tau }\left(s\right) & =M\left(\tilde{\theta }\left(s\right)\right)\ddot{\tilde{\theta }}\left(s\right)+h_{1}\left(\tilde{\theta }\left(s\right)\right)\left[\dot{\tilde{\theta }}\left(s\right)\dot{\tilde{\theta }}\left(s\right)\right]\\ & \;\;\;\;\;\;\;\;\;+h_{2}\left(\tilde{\theta }\left(s\right)\right)\left[{\dot{\tilde{\theta }}\left(s\right)}^{2}\right]+\mu B\dot{\tilde{\theta }}\left(s\right), \end{array}$$
where the tilde ($\tilde{\;}$) means time-normalized, and μ denotes the approximation coefficient that approximately holds when the time dependence of the viscous force is small.

Therefore, the commanded torque change is given as:(9)$$\frac{d\tau (t)}{dt}≃\frac{1}{D^{3}}\frac{d\tilde{\tau }\left(s\right)}{ds}.$$

Substituting the above equation into the cost function of the minimum commanded torque change criterion gives:(10)$$\begin{array}{cc} C_{\tau } & =\sum_{i=1}^{N}\int_{0}^{D}{\left(\frac{d\tau_{i}\left(t\right)}{dt}\right)}^{2}dt, \end{array}$$(11)$$\begin{array}{cc} \; & ≃\sum_{i=1}^{N}\int_{0}^{1}{\left(\frac{1}{D^{3}}\frac{d\tilde{\tau_{i}}\left(s\right)}{ds}\right)}^{2}D\;ds, \end{array}$$(12)$$\begin{array}{cc} \; & ≃\frac{1}{D^{5}}\sum_{i=1}^{N}\int_{0}^{1}{\left(\frac{d\tilde{\tau_{i}}\left(s\right)}{ds}\right)}^{2}ds. \end{array}$$
(13)$$∴\;C_{\tau }≃\frac{a}{D^{5}},\;a=\sum_{i=1}^{N}\int_{0}^{1}{\left(\frac{d\tilde{\tau_{i}}\left(s\right)}{ds}\right)}^{2}ds,$$
where *a* is a constant.

### 2.2. Proof of $C_{\mathrm{err}}≃{\left(W^{⋆}-\gamma /D\right)}^{2}$

The hand endpoint errors Δx(D) and Δy(D) in the *x*- and *y*-axes are expressed by the following equations [23]: (14)$$\begin{array}{cc} \Delta x(D) & ≃\frac{1}{D}\sum_{i=1}^{N}k_{i}\alpha_{i}^{x}, \end{array}$$(15)$$\begin{array}{cc} \alpha_{i}^{x} & =\int_{0}^{1}{\left(J^{x}\left(\tilde{\theta }\left(s\right)\right)B^{-1}\right)}_{i}\left|{\tilde{\tau }}_{i}\left(s,\tilde{\theta }\left(s\right),\dot{\tilde{\theta }}\left(s\right),\ddot{\tilde{\theta }}\left(s\right)\right)\right|z_{i}\left(s\right)\;ds, \end{array}$$(16)$$\begin{array}{cc} \Delta y(D) & ≃\frac{1}{D}\sum_{i=1}^{N}k_{i}\alpha_{i}^{y}, \end{array}$$(17)$$\begin{array}{cc} \alpha_{i}^{y} & =\int_{0}^{1}{\left(J^{y}\left(\tilde{\theta }\left(s\right)\right)B^{-1}\right)}_{i}\left|{\tilde{\tau }}_{i}\left(s,\tilde{\theta }\left(s\right),\dot{\tilde{\theta }}\left(s\right),\ddot{\tilde{\theta }}\left(s\right)\right)\right|z_{i}\left(s\right)\;ds, \end{array}$$
where $k_{i}$ denotes a signal-dependent noise parameter at the *i*-th joint, and $\alpha_{i}^{x}$ and $\alpha_{i}^{y}$ are the time-normalized components, which are calculated based on arm dynamics. $z_{i}\left(s\right)$ is a pseudorandom variable that generates a different sequence every time, according to the standard normal distribution. Thus, $\alpha_{i}^{x}$ and $\alpha_{i}^{y}$ are random variables. The hand endpoint error $\hat{W}$ is defined as the Euclidean distance of Δx(D) and Δy(D) as follows:(18)$$\begin{array}{cc} \hat{W} & \equiv \sqrt{{\Delta x(D)}^{2}+{\Delta y(D)}^{2}}≃\frac{\gamma }{D}, \end{array}$$(19)$$\begin{array}{cc} \gamma & =\sqrt{{\left(\sum_{i=1}^{N}k_{i}\alpha_{i}^{x}\right)}^{2}+{\left(\sum_{i=1}^{N}k_{i}\alpha_{i}^{y}\right)}^{2}}, \end{array}$$
where $\hat{W}$ is approximately represented as a function of *D* with γ as a constant, that is, $\hat{W}$ is inversely proportional to *D*. Consequently, $C_{\mathrm{err}}$ is given as:(20)$$\begin{array}{cc} C_{\mathrm{err}} & ={\left(W^{⋆}-\hat{W}\right)}^{2}, \end{array}$$
(21)$$\begin{array}{cc} \; & ≃{\left(W^{⋆}-\frac{\gamma }{D}\right)}^{2}. \end{array}$$

## 3. Methods

To examine the validity of the movement duration planning model, the behavioral experiment is performed to compare the movement duration estimated by the model with the actual human one. The distribution of the difference between the mean values of the measured and model movement duration was estimated based on Kruschke ’s Bayesian estimation supersedes the t-test (BEST) [24,25].

### 3.1. Subjects

A total of 6 healthy young adults (6 males; age range, 21–24 years) participated in this study. All subjects were right-handed according to the Edinburgh handedness test score (score range, 64.7–100%). Informed consent was obtained from all subjects, and the study was conducted according to the guidelines of the Declaration of Helsinki, and approved by the Ethics Committee of Nagaoka University of Technology (2019–2021 R1-1, 23 August 2019 approved).

### 3.2. Apparatus

The experimental setup is illustrated in Figure 3A. The subject was asked to sit on a chair that was placed facing the table and display (PDP-504P, Pioneer, Tokyo, Japan). The subject placed his right forearm on the table, and the air-sled (a device used to reduce the frictional resistance between the table and the arm) was attached and fixed. Then, the chair height was adjusted such that the table and the subject’s arm were parallel. The chair was moved toward the table until the subject’s chest touched the table. Infra-red markers were affixed to the shoulder, elbow joints, and the hand, and the positions of the three markers were measured at a sampling frequency of 500 Hz using a three-dimensional optical position measurement digitiser (Optotrak Certus, Northern Digital Inc., Waterloo, ON, Canada). The measured current hand positions were projected onto the display installed in front of the subject. A starting point (a circle with a radius of 10.0 mm) and a target (a circle with a radius of 8.00, 15.0, or 25.0 mm) were also shown on the screen, and the subjects performed the experimental task while watching the display.

### 3.3. Task

We sought to investigate how the determined movement duration changes under a different tolerance, starting point, and final point conditions. To this end, the experimental task was designed to have the subject perform reaching movements toward large, medium, and small targets appearing in four directions: Front, back, left, and right, 150 mm away from the starting point. The endpoint accuracy was controlled by changing the tolerance, that is, target size. The target circle radius was 8.00, 15.0, or 25.0 mm. When the experimental program was run, the shoulder positions were defined as the origin of the task coordinate system. The starting circle was displayed in the center of the screen, and the coordinate of the starting circle was defined as a shoulder joint angle of $58^{\circ }$ ($\theta_{1}$), and an elbow joint angle of $96^{\circ }$ ($\theta_{2}$) therefore, the starting coordinates for the right shoulder varied depending on the subject. The targets were placed in four directions: Front ($90^{\circ }$), back ($270^{\circ }$), left ($180^{\circ }$), and right ($0^{\circ }$), based on the starting position. See Figure 3B for details.

There were four directions and three target tolerances thus, a total of 12 types of patterns were displayed on the screen. The pattern of the four directions was pseudo-randomly proposed and switched to the next pattern when the trial was regarded as a success. Each pattern was performed in 32 trials, and the experiment continued until a total of 384 successful trials were conducted.

The subject received the following four instructions:Meet your hand endpoint with the target circle;Avoid corrective movements as much as possible;Increase the number of consecutive successful trials displayed in the lower left of the screen as much as possible;Reach to the target as quickly as possible.

With regard to item 3, we set this instruction to avoid too many failure trails. This also functions to control the success rate, which is defined as the percentage of success among a certain number of trials.

The definition of success was as follows:The hand endpoint is within the target circle;The tangential velocity from the start to the end of the reaching movement is bell-shaped, with one peak.

A corrective movement was considered to have been performed if there was more than one peak and in this case, the trial was regarded as a failure. All trials satisfying item 2 were used in the analysis, that is, even if the hand endpoint was not within the target circle, it was used in the analysis, as long as it was a trial without corrective movement. This was acceptable because the movement duration determined by our proposed model accounts for the movement duration of trials outside the circle, and not the movement duration obtained only including successful trials. This experiment was conducted after the subject had sufficient practice therefore, no clear learning effect was observed in the experimental data.

### 3.4. Data Analysis

#### 3.4.1. Measured Movement Duration

The acquired positional data were low-pass filtered using a third-order, zero-phase-lag Butterworth filter with a cut-off frequency of 10 Hz. The start and end of the movement for each trial were determined based on the tangential velocity, which was calculated using numerical differentiation. The start of the movement was defined as the point following the start cue at which the tangential velocity first exceeded the threshold velocity, defined as 5% of the peak value of the tangential velocity. The end of the movement was defined as the point prior to the end cue at which the tangential velocity fell below the threshold. The time from the start to the end of the movement was then defined as the measured movement duration.

#### 3.4.2. Model Movement Duration

The flow for obtaining the model movement duration $D^{⋆}$ is as follows. First, to calculate the constants *a* and γ, the time-normalized trajectory $\tilde{\theta }\left(s\right)$, μ, and the parameters of arm dynamics and signal-dependent noise were obtained. Subsequently, *a* and γ were calculated, and λ, the weight of the cost function, was determined. Finally, $D^{⋆}$ was determined using the steepest descent method. These details will be explained step-by-step in the next section.

#### 3.4.3. Calculation of *a* and γ

First, the time-normalized trajectory $\tilde{\theta }\left(s\right)$, μ, and the parameters of arm dynamics and signal-dependent noise were obtained to calculate the constants *a* and γ.

The parameters of the arm dynamics consist of the *i*-th link length ($L_{i}$), the distance from the *i*-th joint to the center of mass ($S_{i}$), *i*-th link mass ($m_{i}$), moment of inertia around the *i*-th joint ($I_{i}$), and joint viscosity coefficient responsible for converting the *j*-th joint angular velocity to the *i*-th torque ($B_{ij}$). The $S_{i}$, $m_{i}$, and $I_{i}$ were obtained by the proportional calculation of $L_{i}$ (see [23] for details). $L_{i}$ was calculated from the measured initial hand, elbow, and shoulder positions. $B_{ij}$ was estimated using the method in [4], based on the approximate relationship between the joint torque and the measured viscosity coefficient during static force control. $B_{ij}$ was obtained for each tolerance condition, and then determined for each direction by taking the average over all the tolerance conditions. Table 1 shows the parameters of the subject’s arm dynamics.

The method for estimating the parameters of the signal-dependent noise ($k_{i}$) is described. $k_{i}$ was determined by searching for a pair of $k_{1}$ and $k_{2}$ when the 95% confidence ellipse of the hand endpoints generated by the simulation overlapped the measured one with a grid search algorithm. The hand endpoint error by simulation was obtained by the following procedure (see [23] for details):Joint torque is calculated using measured trajectory data and parameters of arm dynamics.Generate signal-dependent noise at arbitrary $k_{1}$ and $k_{2}$, and add to the joint torque, which is calculated as 1.The hand endpoint error is generated by converting the noise-added torque from the joint space to the task space using the forward dynamics model.

Table 2 shows the obtained signal-dependent noise parameters of the subject.

Once the arm dynamics and noise parameters have been obtained, the time-normalized trajectory can be calculated. As a precondition for calculating the time-normalized trajectory, it is assumed that the human hand path does not change kinematically within a certain movement duration [23]. The time-normalized trajectory was obtained by inputting the movement duration ($D^{\mathrm{tmp}}=1.0\;s$, because of time normalization) and the spatial information of the start and target points to the FIRM using the obtained parameters.

The approximation coefficient μ of Equation (Equation 8) was obtained by the least squares method, where the interval of *D* is defined as [$D^{\mathrm{MIN}}$, $D^{\mathrm{MAX}}$]. The interval of *D* was defined as [0.40, 1.50] s thus, μ=0.5681.

Finally, the constants *a* in Equation (Equation 4) and γ in Equation (Equation 5) were calculated using the above-obtained parameters. The value of γ varies each time, since it includes the pseudo-normal random number *z*. Therefore, we generated 1000 pseudorandom numbers to confirm the probability distribution of γ more accurately.

#### 3.4.4. Determination of λ

The weight λ was determined by the grid search algorithm when the average value of the mean absolute error (MAE) between the average values of the measured and model movement duration was the smallest. The grid search range was from $1\times 10^{5}$ to $1\times 10^{7}$. There are two assumptions about λ: (1) There is one λ per subject, regardless of tasks such as movement direction, and (2) λ exists in each task, that is, each direction. Table 3 shows the λ determined for each direction, and the λ determined as the mean value over the directions based on these assumptions. As shown in Table 3, the determined λ has values of the order of $10^{5}$ and $10^{6}$, and the average value across subjects was $1.28\times 10^{6}$.

#### 3.4.5. Optimization Algorithm

Several engineering methods can be used to obtain the optimal solution (i.e., movement duration *D*) to minimize the cost function. In the current study, the steepest descent method, using the Rprop algorithm [26], was adopted. The Rprop algorithm accelerates the convergence to the minimal solution, while also preventing divergence that can occur with the normal steepest descent method by adjusting the learning rate η according to the sign information of the gradient. The value of η increases slightly as *D* changes in the desired direction if the gradients of the current and next step have the same signs (specifically, η is increased only when the value obtained by multiplying η by the coefficient $\eta^{+}$ is smaller than the threshold $\eta^{\mathrm{MAX}}$ otherwise, η remains unchanged). On the contrary, η is small because the updated *D* is oscillatory if the gradients of the current and next steps have opposite signs (specifically, η is reduced only when the value obtained by multiplying η by the coefficient $\eta^{-}$ is larger than the threshold $\eta^{\mathrm{MIN}}$ otherwise, η remains unchanged). The parameters of the Rprop algorithm are listed in Table 4. The $\eta^{\mathrm{init}}$ denotes the initial value of the learning rate.

#### 3.4.6. Bayesian Estimation of the Difference between Two Mean Values

The distribution of the difference between the mean values of the measured and model movement duration was estimated based on Kruschke (2013, 2014)’s Bayesian estimation supersedes the *t*-test (BEST) [24,25]. We used the MATLAB Toolbox for Bayesian Estimation [27] which is a MATLAB implementation of Kruschke’s R code BEST package. In the posterior distribution estimation of the difference between the mean values, it is important that 95% HDI (highest density interval) is included in ROPE (region of practical equivalence). If the effect size is within ROPE, two mean values are considered to be equal. The ROPE was set to [−0.1, 0.1]. For the prior distribution, a non-information prior distribution (uniform distribution) was used. The parameters used in the analysis are as follows:Number of separate Markov chain Monte Carlo methods (MCMC) chains: 3;Number of MCMC steps that are saved for each chain: 4000;Number of steps that are thinned: 5.

## 4. Results

In the current study, we investigated the following: (1) How closely the movement duration determined by the proposed model is to the actual human movement, and (2) how much the trajectory generated by inputting the movement duration obtained by the model into the FIRM can reproduce the features of human arm kinematics. For (1), the distribution of the difference between the mean values of the measured and model movement duration was estimated using Bayesian statistics. If zero is included in the 95% HDI of the posterior distribution, it can be said that there is no difference between the average values with a 95% probability. The movement duration and hand trajectory were obtained by the procedure shown in Figure 1. The parameters used for the numerical calculations were as follows:Parameters of the arm dynamics shown in Table 1 were usedThe parameters of signal-dependent noise shown in Table 2 were used;μ, calculated with the interval of *D* [0.40, 1.50] s, was 0.5681;λ, as shown in Table 3;The sampling frequency was 500 Hz;The number of generations of pseudo-normal random numbers was 1000.

### 4.1. Comparison of Measured and Model Movement Durations

Figure 4 shows the comparison results between the movement duration measured in the behavioral experiment (black box plot) and the movement duration determined by the model (red box plot) in all subjects. Similar to the measured movement duration, the model movement duration became longer as the tolerance $W^{⋆}$ became smaller. By considering the above, it can be said that the model represents the speed-accuracy trade-off that appears in the measured data. At 8.00, 15.0, and 25.0 mm of tolerance, the model could represent the tendency of the measured data for the left to have the longest movement duration and back to have the shortest movement duration among the four directions. These results were similar in most subjects.

Figure 5 shows the mean absolute error (MAE) between the average values of the measured and model movement duration over the subjects. The MAE between them was about 0.2 s at the maximum. As an overall trend, the backward movement had the lowest MAE, and the leftward movement had the highest MAE among the four directions. The case where the model parameter λ was determined for each direction (Figure 5A) and the case where λ, which is the average of them, was used in each direction (Figure 5B) are shown. When λ was determined for each direction, the model movement duration was a little closer to that of the actual duration. Even if one λ was determined in four directions, the error between the model and human did not become large. It is noteworthy from Figure 5B that this difference in each direction is not adjusted by λ.

In conclusion, the model movement duration was close to the human movement duration, and the speed-accuracy trade-off shown in the measured data was also expressed by the model.

Table 5 shows the Bayesian estimation result of the difference between the average value of the measured and model movement duration for all subjects. Specifically, the 95%HDI and %inROPE of the posterior distribution of the difference between them, that is, the rate of posterior probabilities within the ROPE were shown. The %inROPE shown in bold shows that the effect size is within ROPE [−0.1,0.1], and it could be confident that there is no difference between the average values of the measured movement duration and the model movement duration. Based on this result, it could be said that the average values of the measured and model movement duration were substantially equal under the condition of about one-third of the whole.

### 4.2. Comparison of Measured and Model Kinematics

Figure 6 shows the comparison between the measured and model results of the hand path and tangential velocity. As shown in Figure 6A, the trajectories obtained by the FIRM reproduced the features of the human hand path, which is a slightly curved, but almost straight path. In addition, the kinematic characteristics of the trajectory were hardly changed with the change in the given tolerance condition, in both the measured and model conditions, that is, the trajectory form only slightly changed, even if the movement duration changed. As shown in Figure 6B, the tangential velocity obtained by the FIRM showed a unimodal bell shape similar to that of human motion. Furthermore, the tangential velocity was scaled with the change in movement duration *D*. These tendencies were similar in other subjects as well as in the representative subject (Appendix A).

We have confirmed that the above results coincide with [15]. Next, we focused on not only the average movement duration, but also on the distribution of movement duration. As can be seen from the tolerance of 8.00 mm in Figure 6B (left-most column), the measured and model average movement durations for each movement direction were fairly close, but the model peak velocities were located slightly behind those measured in all cases. In addition, the model movement duration appeared to be more variable than the measured one at the endpoint of velocity (i.e., near the movement duration).

In any case, regarding the average value, the hand paths and tangential velocities obtained by the FIRM capture the features of the human reaching movement well.

## 5. Discussion

We propose that the movement duration is determined by a cost function that considers two things: To get smoother motor commands and to fit the hand endpoint variability in a given tolerance. Here, the movement duration optimization module was added to the FIRM, which is a computational model of human arm movement. The movement duration determined by the proposed model represented a universal phenomenon, such as the features of hand trajectory and velocity and the speed-accuracy trade-offs found in actual human movement (Figure 4). The MAE between the average values of the measured and model movement duration was about 0.2 s at the maximum (Figure 5). Furthermore, from the Bayesian estimation result of the difference between these mean values, it could be said that the average values of the measured and model movement duration were substantially equal under the condition of about one-third of the whole (Table 5). The trajectory generated by the FIRM with this movement duration as the input reproduced the actual trajectory well (Figure 6).

As mentioned in Section 4.2, it is important to be able to reproduce not only the average value of the movement duration, but also its distribution with a model. Figure 7 shows the distribution of the measured and model movement duration shown in Figure 4A. The observed distribution and model distribution were not similar; the distribution of observed movement duration was likely to be normal, whereas the distribution of movement duration determined by the model was likely to be exponential distribution. In Equations (Equation 15) and (Equation 17), Δx(D) and Δy(D) follow a normal distribution, with an average of 0, such that the Euclidean distance γ follows a Rayleigh distribution therefore, $\hat{W}$ in Equation (Equation 19) also follows the Rayleigh distribution. It is considered that the movement duration $D^{⋆}$ became an exponential because the $D^{⋆}$ is obtained using Equation (Equation 19). These findings suggest that further research is necessary to reproduce the distribution of actual human movement duration using the model.

As shown in Figure 5, the leftward movement had the highest MAE among the four directions. Such an error is mainly due to the error caused by making some approximations and assumptions in modeling. Therefore, the error should be reduced by considering the approximation and assumptions more strictly in future work.

We investigated how the $W^{⋆}$–*D* profile behaves when parameter λ is changed. As shown in Equation (Equation 1), λ is a parameter indicating the extent to which the tolerance term should be emphasized with respect to the smoothness term. λ was changed to ×0.5, ×1.0, ×2.0, and ×4.0, and the base value of λ is shown in Table 3. Figure 8 shows that $C_{\mathrm{err}}$ is more important than $C_{\tau }$ when λ is increased thus, the movement duration becomes shorter.

One of the limitations of this study is the number of subjects. There are few subjects (six subjects) therefore, we could not rely on the power analysis, which requires a large number of samples. Instead, Bayesian statistics were used to estimate the difference in mean values.

Unfortunately, there is currently no clear rationale as to why the movement duration does not become too slow. If the shortest movement duration was determined under a certain condition, we found a sigmoidal shape of the CoT, as described by [18] (the $W^{⋆}\geq \hat{W}$ area of the green dashed line in Figure 2). Our findings represent the first step in clarifying the information processing mechanism of the brain during human-reaching movements.

As shown in Figure 4, Figure 5, Figure 6 and Figure 7, FIRM does reproduce the human motor trajectory well, but at the same time shows that the present distribution still needs improvement. This is the first time that FIRM has been extended to a stochastic model, and these will need further research. FIRM can be realized with a biologically plausible neural network model [14,15]. Therefore, it can be expected that a model incorporating this movement duration planning module into the FIRM is closer to the calculation performed in the human brain. We have incorporated the cost of exercise time to further improve this superior model. As explained in the Introduction section, it is the same idea as the model of Berret et al. (2016) in terms of CoT. In future, we would like to consider the possibility of multiple costs that incorporate time costs into FIRM. The possibility of multiple costs that incorporate CoT into FIRM will be considered in future.

## 6. Conclusions

We proposed a model for trajectory planning that incorporates movement duration planning into the FIRM. The two constants *a* and γ were calculated based on the human arm dynamics model in the presence of the signal-dependent noise. Movement duration was estimated by minimizing the cost function that considers the smoothness of motor commands and endpoint tolerance.

Hereafter, in modeling the movement duration planning, the hardware-implementation level research is expected to progress as well as the computational theory, representation, and algorithm level research. We further need to consider the possible costs that can explain why we do not move slower.

## Figures and Tables

**Figure 1 brainsci-11-00149-f001:**
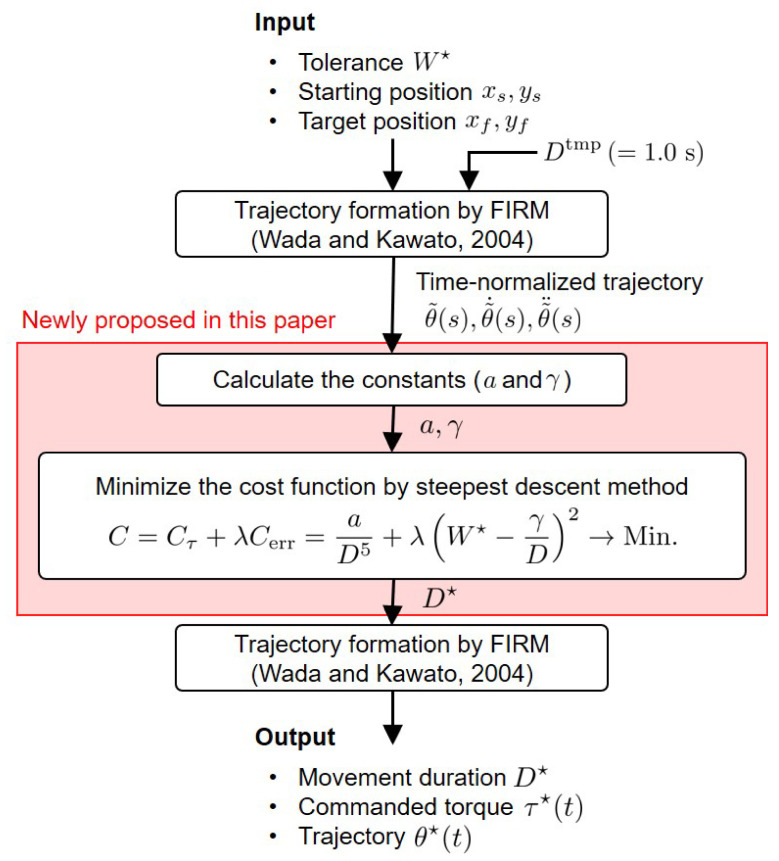
Schematic diagram of proposed computational models in point-to-point human arm reaching movement. The parts newly proposed in this paper are shown in a red box.

**Figure 2 brainsci-11-00149-f002:**
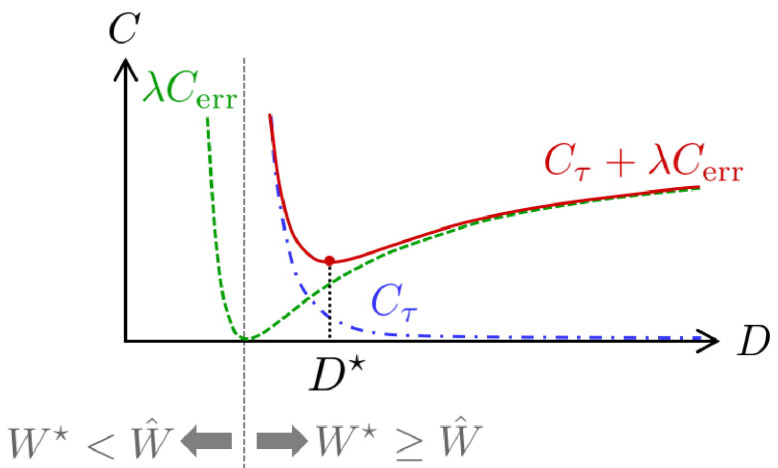
Illustration of the proposed cost function. The blue dash-dotted line is the trajectory cost, i.e., the cost of the minimum commanded torque change ($C_{\tau }$). The green dashed line is the cost of time (CoT), i.e., the squared error between tolerance and estimated hand endpoint variability ($C_{\mathrm{err}}$). The red solid line is the weighted sum of the trajectory cost and the CoT ($C_{\tau }+\lambda C_{\mathrm{err}}$). As mentioned in [18], infinitely slow movements are considered to be optimal for the trajectory costs without the CoT.

**Figure 3 brainsci-11-00149-f003:**
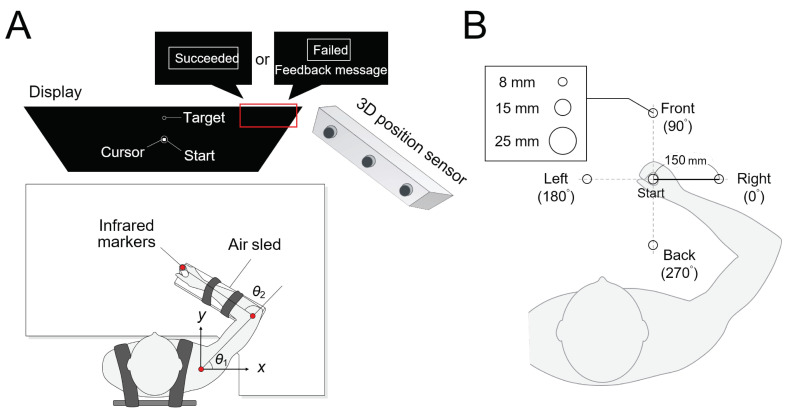
Top view of experimental setup. (**A**) Experimental environment, showing definitions of joint angles $\theta_{1}$ and $\theta_{2}$ and *x*– and *y*–coordinates; similar to our previous study [23]. (**B**) Conditions of three different target sizes and the placements of four target positions with respect to the central circle.

**Figure 4 brainsci-11-00149-f004:**
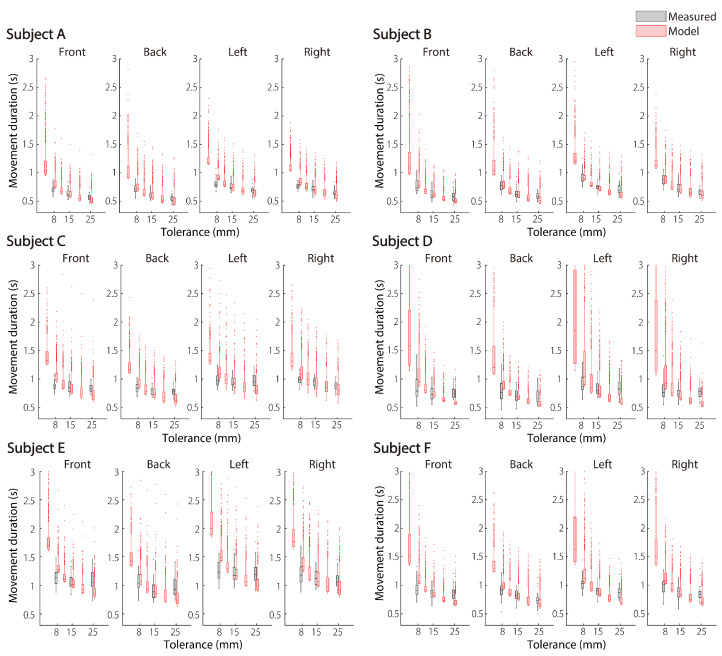
Comparison of the measured and model movement durations for all subjects. When the λ determined for each direction was used. The horizontal axis represents tolerance, and the vertical axis represents movement duration. The black box plot denotes the measured movement duration at a given tolerance: 8.00, 15.0, and 25.0 mm. The red box plot represents the model movement duration generated for 1000 trials when tolerances of 4.00, 8.00, 11.5, 15.0, 20.0, and 25.0 mm were given. The light red dots indicate outliers (greater than third quartile + 1.5 × IQR).

**Figure 5 brainsci-11-00149-f005:**
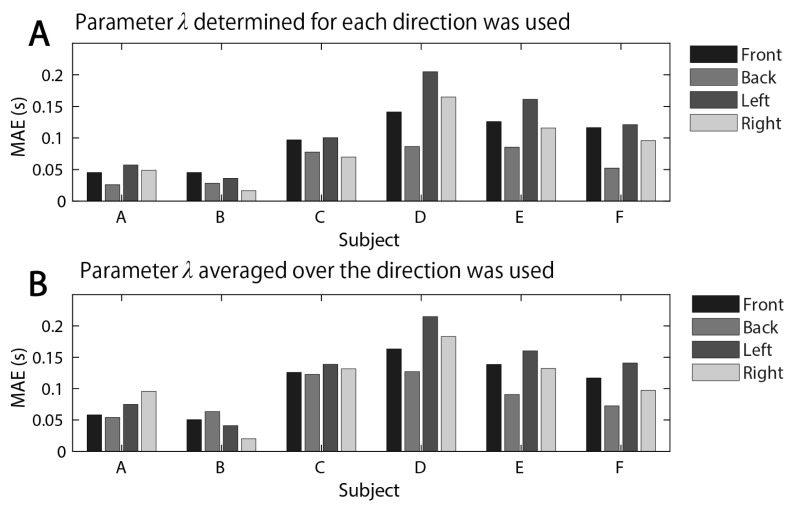
The MAE (mean absolute error) between the average values of the measured and model movement duration over the subjects. (**A**) When the λ determined for each direction was used. (**B**) When the average value of λ over four directions was used.

**Figure 6 brainsci-11-00149-f006:**
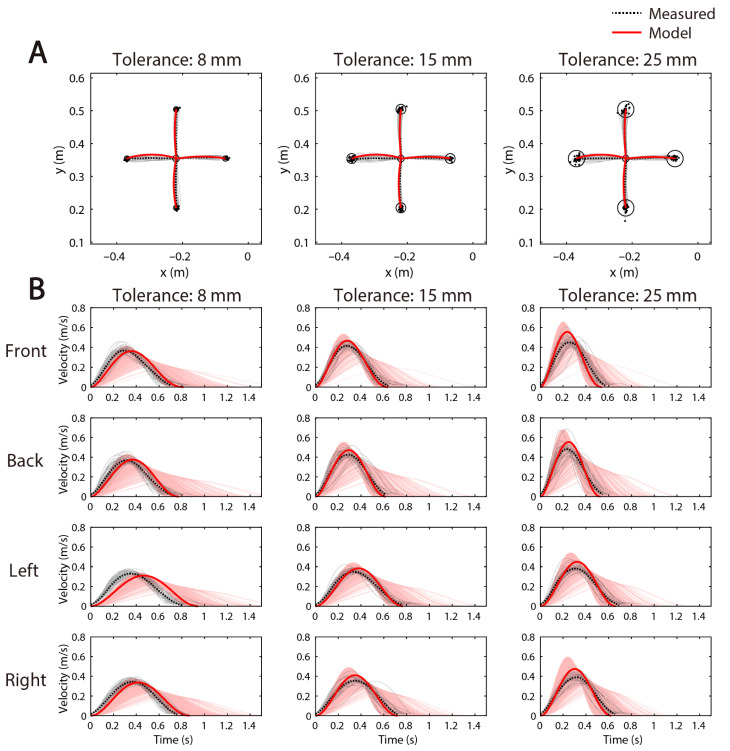
Comparison of the measured and model kinematics of a representative subject (subject A). (**A**) the measured and model hand paths, (**B**) the measured and model tangential velocities. For both (**A**,**B**), the λ determined for each direction was used. The three axes arranged in a row shows the tolerance: 8.00, 15.0, and 25.0 mm, respectively. The light solid black and dashed black lines represent the measured one for each trial and the average path over all the trials, respectively. The red solid line denotes the average one over all the trials. The light red line obtained by inputting the movement duration determined by the proposed model into the forward inverse relaxation model (FIRM) for each trial; however, this is obscured by the red solid line in (**A**). The black dot represents the endpoint of the hand, and the circle represents the area of tolerance.

**Figure 7 brainsci-11-00149-f007:**
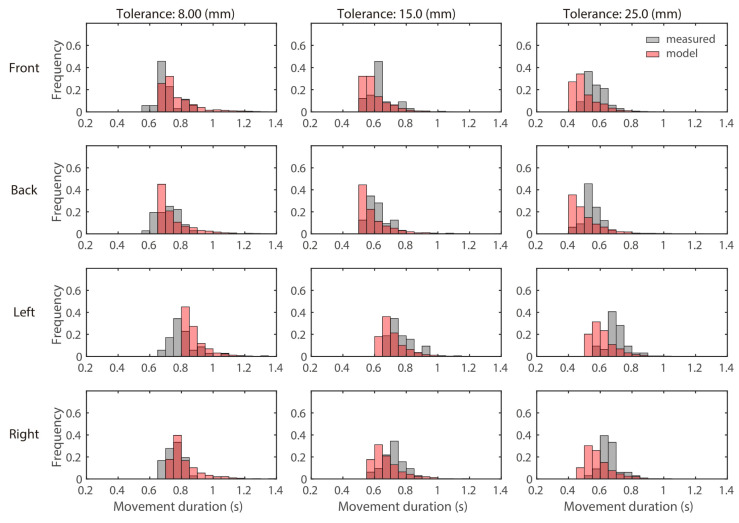
Comparison between the distributions of the measured and model movement durations. The results of a representative subject (subject A) are shown. The λ determined for each direction was used. The histograms are normalized such that the total height of all bars is 1.

**Figure 8 brainsci-11-00149-f008:**
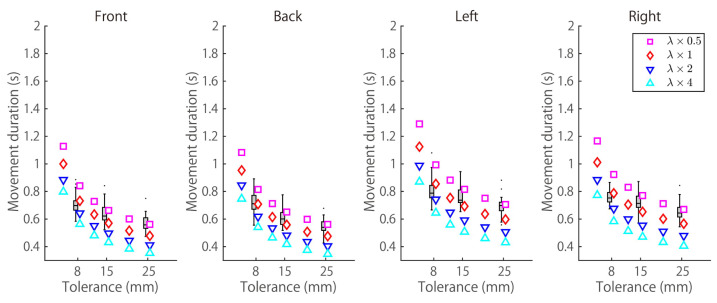
Median movement duration profiles obtained by the model when λ is changed in ×0.5, ×1, ×2, and ×4 (the base values of λ are shown in Table 3). The results of a representative subject (subject A) are shown.

**Table 1 brainsci-11-00149-t001:** Parameters of the arm dynamic model. $L_{1}$ and $L_{2}$ are the length of the upper arm and forearm, respectively. $S_{1}$ and $S_{2}$ are the length from the shoulder and elbow joints to the center of gravity, respectively. $m_{1}$ and $m_{2}$ are the masses of the upper arm and forearm, respectively. $I_{1}$ and $I_{2}$ are the moments of inertia around the shoulder and elbow joints, respectively. $B_{11}$, $B_{12}$, $B_{21}$, and $B_{22}$ are the viscosity coefficient from the joint angular velocity of the right number subscript to the joint torque of the left number subscript. The viscosity coefficient was calculated for each movement direction.

Subject	A	B	C	D	E	F	Mean±SD *
$L_{1}$ (m)	0.281	0.261	0.296	0.326	0.273	0.260	0.283 ± 0.025
$L_{2}$ (m)	0.351	0.323	0.329	0.335	0.316	0.305	0.327 ± 0.016
$S_{1}$ (m)	0.105	0.097	0.111	0.124	0.102	0.096	0.106 ± 0.010
$S_{2}$ (m)	0.171	0.158	0.161	0.164	0.155	0.150	0.160 ± 0.007
$m_{1}$ (kg)	1.390	1.284	1.472	1.636	1.348	1.276	1.401 ± 0.136
$m_{2}$ (kg)	1.871	1.788	1.806	1.825	1.768	1.736	1.799 ± 0.047
$I_{1}$ $\left(\mathrm{kg}\cdot m^{2}\right)$	0.024	0.019	0.028	0.039	0.022	0.018	0.025 ± 0.008
$I_{2}$ $\left(\mathrm{kg}\cdot m^{2}\right)$	0.049	0.039	0.041	0.043	0.037	0.034	0.041 ± 0.005
Front							
$B_{11}$ $\left(\mathrm{kg}\cdot m^{2}/s\right)$	0.653	0.658	0.651	0.658	0.644	0.651	0.652 ± 0.005
$B_{12}$ $\left(\mathrm{kg}\cdot m^{2}/s\right)$	0.180	0.178	0.177	0.177	0.176	0.177	0.178 ± 0.001
$B_{21}$ $\left(\mathrm{kg}\cdot m^{2}/s\right)$	0.180	0.178	0.177	0.177	0.176	0.177	0.178 ± 0.001
$B_{22}$ $\left(\mathrm{kg}\cdot m^{2}/s\right)$	0.784	0.776	0.769	0.771	0.766	0.767	0.772 ± 0.007
Back							
$B_{11}$ $\left(\mathrm{kg}\cdot m^{2}/s\right)$	0.636	0.643	0.642	0.647	0.638	0.642	0.642 ± 0.004
$B_{12}$ $\left(\mathrm{kg}\cdot m^{2}/s\right)$	0.182	0.181	0.179	0.180	0.177	0.178	0.180 ± 0.002
$B_{21}$ $\left(\mathrm{kg}\cdot m^{2}/s\right)$	0.182	0.181	0.179	0.180	0.177	0.178	0.180 ± 0.002
$B_{22}$ $\left(\mathrm{kg}\cdot m^{2}/s\right)$	0.794	0.789	0.779	0.785	0.772	0.776	0.782 ± 0.008
Left							
$B_{11}$ $\left(\mathrm{kg}\cdot m^{2}/s\right)$	0.689	0.679	0.667	0.684	0.652	0.663	0.672 ± 0.014
$B_{12}$ $\left(\mathrm{kg}\cdot m^{2}/s\right)$	0.177	0.177	0.176	0.177	0.176	0.177	0.177 ± 0.001
$B_{21}$ $\left(\mathrm{kg}\cdot m^{2}/s\right)$	0.177	0.177	0.176	0.177	0.176	0.177	0.177 ± 0.001
$B_{22}$ $\left(\mathrm{kg}\cdot m^{2}/s\right)$	0.770	0.771	0.766	0.770	0.764	0.769	0.768 ± 0.003
Right							
$B_{11}$ $\left(\mathrm{kg}\cdot m^{2}/s\right)$	0.684	0.675	0.664	0.684	0.651	0.660	0.670 ± 0.014
$B_{12}$ $\left(\mathrm{kg}\cdot m^{2}/s\right)$	0.177	0.177	0.176	0.177	0.176	0.177	0.177 ± 0.000
$B_{21}$ $\left(\mathrm{kg}\cdot m^{2}/s\right)$	0.177	0.177	0.176	0.177	0.176	0.177	0.177 ± 0.000
$B_{22}$ $\left(\mathrm{kg}\cdot m^{2}/s\right)$	0.768	0.770	0.767	0.770	0.764	0.769	0.768 ± 0.002

* SD: Standard deviation across subjects.

**Table 2 brainsci-11-00149-t002:** Parameters of signal-dependent noise. $k_{1}$ and $k_{2}$ are coefficients that determine the magnitude of signal-dependent noise, and indicates the effect of the commanded torque of the subscript number on noise.

Subject	A	B	C	D	E	F	Mean±SD
$k_{1}$ (shoulder)	0.180	0.300	0.100	0.710	0.530	0.660	0.413 ± 0.256
$k_{2}$ (elbow)	0.660	0.740	0.590	1.070	0.740	0.720	0.753 ± 0.166

**Table 3 brainsci-11-00149-t003:** Parameter λ used in the movement duration optimization model for each subject.

Subject	A	B	C	D	E	F	Mean
Front	$1.44\times 10^{6}$	$2.03\times 10^{6}$	$5.82\times 10^{5}$	$9.37\times 10^{5}$	$1.60\times 10^{5}$	$6.18\times 10^{5}$	$9.61\times 10^{5}$
Back	$1.24\times 10^{6}$	$1.15\times 10^{6}$	$5.49\times 10^{5}$	$6.39\times 10^{5}$	$2.14\times 10^{5}$	$3.77\times 10^{5}$	$6.95\times 10^{5}$
Left	$2.58\times 10^{6}$	$2.12\times 10^{6}$	$1.61\times 10^{6}$	$3.01\times 10^{6}$	$1.90\times 10^{5}$	$8.27\times 10^{5}$	$1.72\times 10^{6}$
Right	$3.05\times 10^{6}$	$1.88\times 10^{6}$	$1.62\times 10^{6}$	$3.13\times 10^{6}$	$2.31\times 10^{5}$	$5.57\times 10^{5}$	$1.74\times 10^{6}$
Mean	$2.08\times 10^{6}$	$1.79\times 10^{6}$	$1.09\times 10^{6}$	$1.93\times 10^{6}$	$1.99\times 10^{5}$	$5.95\times 10^{5}$	$1.28\times 10^{6}$

**Table 4 brainsci-11-00149-t004:** Parameters of the steepest descent method with the Rprop algorithm.

$\eta^{\mathrm{init}}$	$\eta^{+}$	$\eta^{-}$	$\eta^{\mathrm{MAX}}$	$\eta^{\mathrm{MIN}}$
0.010	1.20	0.50	50.0	0.000001

**Table 5 brainsci-11-00149-t005:** Bayesian statistics results for the difference between the mean of the measured and model movement duration for all subjects. The 95% HDI (highest density interval) is an interval which the true value of parameter exist with 95% probability. In this case, we focus on whether the 95% HDI contains zeros. ROPE (region of practical equivalence) was set to [−0.1,0.1]. %inROPE means the percentage of the posterior probability mass within the ROPE. The bold text denotes ≥1%inROPE.

		Subject A		Subject B		Subject C	
		**95%HDI**	**%inROPE**	**95%HDI**	**%inROPE**	**95%HDI**	**%inROPE**
Front	8 mm	[−0.087,−0.05]	0%	[−0.067,−0.008]	**2%**	[−0.151,−0.099]	0%
	15 mm	[0.007,0.043]	**1%**	[−0.006,0.054]	**7%**	[−0.002,0.086]	**7%**
	25 mm	[0.0367,0.077]	0%	[0.062,0.114]	0%	[0.093,0.151]	0%
Back	8 mm	[−0.022,0.028]	**27%**	[−0.021,0.046]	**21%**	[−0.077,−0.008]	**2%**
	15 mm	[0.016,0.061]	0%	[0.002,0.053]	**3%**	[0.034,0.094]	0%
	25 mm	[0.039,0.075]	0%	[0.061,0.105]	0%	[0.123,0.174]	0%
Left	8 mm	[−0.139,−0.090]	0%	[−0.016,0.047]	**16%**	[−0.154,−0.078]	0%
	15 mm	[−0.010,0.041]	**20%**	[0.027,0.064]	0%	[0.003,0.070]	**5%**
	25 mm	[0.034,0.075]	0%	[0.064,0.124]	0%	[0.128,0.211]	0%
Right	8 mm	[−0.092,−0.055]	0%	[−0.003,0.064]	**7%**	[−0.139,−0.080]	0%
	15 mm	[0.003,0.051]	**4%**	[−0.020,0.047]	**26%**	[−0.006,0.054]	**14%**
	25 mm	[0.026,0.065]	0%	[0.029,0.074]	0%	[0.0050,0.103]	0%
		**Subject D**		**Subject E**		**Subject F**	
		**95%HDI**	**%inROPE**	**95%HDI**	**%inROPE**	**95%HDI**	**%inROPE**
Front	8 mm	[−0.218,−0.106]	0%	[−0.179,−0.083]	0%	[−0.185,−0.110]	0%
	15 mm	[−0.018,0.069]	**13%**	[−0.022,0.062]	**22%**	[0.007,0.060]	**2%**
	25 mm	[0.129,0.210]	0%	[0.137,0.264]	0%	[0.116,0.195]	0%
Back	8 mm	[−0.145,−0.036]	0%	[−0.079,0.022]	**15%**	[−0.097,−0.012]	**1%**
	15 mm	[−0.009,0.074]	**8%**	[−0.035,0.084]	**23%**	[0.008,0.088]	**2%**
	25 mm	[0.072,0.172]	0%	[0.128,0.253]	0%	[0.024,0.097]	0%
Left	8 mm	[−0.290,−0.171]	0%	[−0.294,−0.168]	0%	[−0.101,−0.033]	0%
	15 mm	[0.028,0.106]	0%	[−0.039,0.072]	**24%**	[0.027,0.083]	0%
	25 mm	[0.188,0.298]	0%	[0.143,0.280]	0%	[0.138,0.216]	0%
Right	8 mm	[−0.173,−0.092]	0%	[−0.191,−0.059]	0%	[−0.116,−0.035]	0%
	15 mm	[0.036,0.091]	0%	[−0.001,0.117]	**6%**	[0.022,−0.106]	0%
	25 mm	[0.194,0.280]	0%	[0.101,0.207]	0%	[0.127,0.185]	0%

## Data Availability

The data presented in this study are available on request from the corresponding author.

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
