# Peer review of "Forward Inverse Relaxation Model Incorporating Movement Duration Optimization"

_brainsci, 2021, doi:10.3390/brainsci11020149_

Round 1
Reviewer 1 Report
In this paper, the Authors propose a FIRM model to represent natural movements of human arms.
The work is generally well structured, but it needs some revision
- I am not an English mother tongue researcher, but some syntactic structures seemed strange. Please, submit your article to a professional English proof-reader.
- page 1, lines 19-21: “To examine how humans can perform skilled movement, we first considered reaching movements on a horizontal plane, which represent relatively simple human movements to measure using a computational approach”: this is a very strange sentence to begin the introductory section of an article. I would expect a more general introduction, and at the end of it, or even better, in the methods section, the description of which actions have been taken into consideration in this research work.
- page 6, lines 106 – 110: Have you computed a power analysis on your data? Six participants is an extremely low number of subjects. If your study is underpowered, please consider to collect more data.
- Willmott and Matsuura(2005) have suggested that the RMSE is not a good indicator of average model performance and might be a misleading indicator of average error. The mean absolute error (MAE) is a better index for this purpose. Please, consider to add also the MAE to your analysis.
Willmott, C. and Matsuura, K.: Advantages of the Mean Absolute Error (MAE) over the Root Mean Square Error (RMSE) in assessing average model performance, Clim. Res., 30, 79–82,2005
Author Response
We appreciate the valuable comments provided by the reviewers to further improve our manuscript.
In accordance with these comments, we have revised our manuscript; the changes in the revised manuscript are shown in red.
We would like to mention that the primary purpose of our study was to elucidate the information processing mechanism of the human brain, especially during movement duration planning. Hence, we proposed a model for trajectory planning that incorporates movement duration planning that considers the smoothness of motor commands and endpoint tolerance. Although the FIRM is an existing model, the module of the movement duration planning algorithm is the novelty of this research. The movement durations, hand paths, and velocity profiles estimated by our model well represented the features of human reaching movement. This is very important as it may explain motor information processing in the human brain. We have modified the manuscript to make our purpose more clear. We believe that our revisions have addressed all the reviewers’ comments and will satisfy them and readers of the manuscript.

Reviewer 2 Report
In this study from Takeda et al a new framewok called the forward inverse relation model (FIRM) is presented. From the current version of the paper, it is not clear to the reviewer what aspects of the algorithm are new to the (Wada and Kawato, 2004) and the own cited papers in (Wada et al 1993, 1995). I have several major concerns in the current version of the manuscript. I will point them below based on each section:
Introduction:
- What are the new alogirthms changes in this paper to the previously published papers from the authors themselves?
- The authors claim in the introduction that they demonstrate the movement duration determined by the proposed model is close to the true human movement. Please explain how?
Movement duration optimization module:
- Based on Figure 1, if the main step in the algorithm like the trajectory formation and the authors cite (Wada and Kawato 2004) then the steps of calcualting the constants and minimising the cost function is new in this paper?
Methods:
- In my view, the sample size is too low of n = 6 for the conclusions in the mansuscript. Either the authors need to do a power calculation to to justify the sample size or to increase the sample size.
- A figure on the experimental setup would be easier for the reviewer to follow the experiments.
Data Analysis:
- Explain all the parameters and the information in Table 1 and Table 2 with a detailed legend. It is not easy to follow in the present version of the mansucript.
Results:
- The comparison of measured and model movement durations and the measured and model kinematics needs ststistics to justify the conclusion from the authors. At present the RMSE only gives you a subjective value on each individual subject. A quatitative way would be to do for ex: Bayesian statistics (https://link.springer.com/article/10.3758/s13423-017-1272-1) on the data obtained from the six subhjects to the model data.
- The number of subjects are too few for any other non-parametric statistics instead the statistics in the distribution would be good.
Discussion:
- The authors discuss for the distribution further research is needed and claim the FRIM estimation is closer to the calculation performed in the human brain in my view is far-fethched.
- The format of the discussion is not clear, there is currently no discussion about existing literature in this topic and how the authors model is better than the existing ones instead they only discuss the resutls and the perspectives of their current work.
Figures:
Why the figures 3 to 7 or only from a representative subject. The authors need to show the overall results of all the included subjects
Author Response
We appreciate the valuable comments provided by the reviewers to further improve our manuscript.
In accordance with these comments, we have revised our manuscript; the changes in the revised manuscript are shown in red.
With respect to comments #1 by reviewer #2, we would like to mention that the primary purpose of our study was to elucidate the information processing mechanism of the human brain, especially during movement duration planning. Hence, we proposed a model for trajectory planning that incorporates movement duration planning that considers the smoothness of motor commands and endpoint tolerance. Although the FIRM is an existing model, the module of the movement duration planning algorithm is the novelty of this research. The movement durations, hand paths, and velocity profiles estimated by our model well represented the features of human reaching movement. This is very important as it may explain motor information processing in the human brain. We have modified the manuscript to make our purpose more clear. We believe that our revisions have addressed all the reviewers’ comments and will satisfy them and readers of the manuscript.

Round 2
Reviewer 2 Report
I have now looked at the revisions in the manuscript and the point-to-point reply to my comments. In my view the authors have a done great job to answer all my comments. I just have one minor comment.
The authors needs to write in few sentences after the discussion the main conclusion of the study. Fro ex: the two constants were estimated in the FIRM model and able to predict the human arm movement...and so on
Author Response
Point-by-Point Response to Reviewers
We appreciate the valuable comments provided by the reviewers to further improve our manuscript.
In accordance with these comments, we have revised our manuscript; the changes in the revised manuscript are shown in red.
Reviewer #2
Comments and Suggestions for Authors:
I have now looked at the revisions in the manuscript and the point-to-point reply to my comments. In my view the authors have a done great job to answer all my comments. I just have one minor comment.
The authors needs to write in few sentences after the discussion the main conclusion of the study. For ex: the two constants were estimated in the FIRM model and able to predict the human arm movement...and so on
Author's Notes to Reviewer:
Thank you for your suggestion. We have added some sentences based on your comments in Conclusion section after the Discussion (p. 14, lines 366-373). We believe that the manuscript has become more cohesive with this modification.
Again, thank you for giving us the opportunity to strengthen our manuscript with your valuable comments and queries. We have worked hard to incorporate your feedback and hope that these revisions persuade you to accept our submission.
